# A Dark Septate Endophyte Improves Cadmium Tolerance of Maize by Modifying Root Morphology and Promoting Cadmium Binding to the Cell Wall and Phosphate

**DOI:** 10.3390/jof9050531

**Published:** 2023-04-29

**Authors:** Si Chen, Guangqun Zhang, Xinran Liang, Lei Wang, Zuran Li, Yongmei He, Bo Li, Fangdong Zhan

**Affiliations:** 1College of Resources and Environment, Yunnan Agricultural University, Kunming 650201, China; 2College of Horticulture and Landscape, Yunnan Agricultural University, Kunming 650201, China

**Keywords:** dark septate endophytes, cadmium stress, root morphology, subcellular distribution, chemical forms

## Abstract

Dark septate endophytes (DSEs) can improve the performance of host plants grown in heavy metal-polluted soils, but the mechanism is still unclear. A sand culture experiment was performed to investigate the effects of a DSE strain (*Exophiala pisciphila*) on maize growth, root morphology, and cadmium (Cd) uptake under Cd stress at different concentrations (0, 5, 10, and 20 mg·kg^−1^). The results indicated that the DSE significantly improved the Cd tolerance of maize, causing increases in biomass, plant height, and root morphology (length, tips, branch, and crossing number); enhancing the Cd retention in roots with a decrease in the transfer coefficient of Cd in maize plants; and increasing the Cd proportion in the cell wall by 16.0–25.6%. In addition, DSE significantly changed the chemical forms of Cd in maize roots, resulting in decreases in the proportions of pectates and protein-integrated Cd by 15.6–32.4%, but an increase in the proportion of insoluble phosphate Cd by 33.3–83.3%. The correlation analysis revealed a significantly positive relationship between the root morphology and the proportions of insoluble phosphate Cd and Cd in the cell wall. Therefore, the DSE improved the Cd tolerance of plants both by modifying root morphology, and by promoting Cd binding to the cell walls and forming an insoluble phosphate Cd of lower activity. These results of this study provide comprehensive evidence for the mechanisms by which DSE colonization enhances Cd tolerance in maize in root morphology with Cd subcellular distribution and chemical forms.

## 1. Introduction

Soil pollution by heavy metals is a significant global problem that has threatened sustainable development in recent years [1]. Anthropogenic activities such as metal mining and sewage irrigation, massive discharges of industrial effluent, and the application of fertilizers accelerate heavy metal contamination in soils [2,3]. Cadmium (Cd) is considered to be one of the most toxic heavy metals in the environment [4]. Although Cd is usually present in trace amounts, its high toxicity, nonbiodegradability, and persistence pose a considerable threat to the environment and human health through the food chain [1,5]. Notably, excessive Cd concentrations within metabolically active organs can cause severe physiological and biochemical toxicity to most plant, such as cell membrane permeability and plant protection enzyme systems, thereby inhibiting plant growth [6,7]. Therefore, it is necessary to efficiently improve plant tolerance and to take ecological security measures to reduce the accumulation of Cd in plants. 

The root system is the main organ of plants for nutrient and water uptake, and it is affected by both beneficial and toxic environmental elements [8]. Studies have found the association between root traits and Cd stress. The root system responds to Cd-induced stress by adjusting its morphological structure and root growth of the plant (e.g., root elongation, length, surface area, and biomass) [9,10]. Furthermore, antagonism between Cd and mineral nutrients can lead to nutrient deficiencies, which inhibit plant growth [11]. The subcellular distribution of Cd in plants is critical in determining the degree of Cd toxicity and tolerance mechanisms [12]. The cell wall is the first barrier against Cd ions entering the cytoplasm and the leading site for Cd detoxification [13]. The detoxification system is highly dependent on functional groups in the cell wall (e.g., carboxyl, hydroxyl, and amino, etc.), which act as negatively charged sites to bind to Cd ions and regulate the transmembrane transport of Cd [14]. Simultaneously, different chemical forms of Cd extracted with different extractants (such as 80% ethanol, deionized water, 1 M NaCl, 2% acetic acid, and 0.6 M HCl) have additional toxicity and migration capacity in plant cells [15]. For example, insoluble phosphate–Cd complex precipitates (extracted with 2% acetic acid) have lower in vivo mobility and toxicity, and play an essential role in improving the tolerance of plants to stressful environments [16,17].

Dark septate endophytes (DSEs) are one of the most studied groups of fungal endophytes that colonize plant roots in stressful environments [18,19,20]. Particularly, DSEs have obvious dematiaceous septate hyphae and melanized microsclerotia, mainly composed of phenolic compounds formed by oxidation, which can enhance the tolerance of host plants and play multifunctional roles for plant growth in heavy metal-polluted ecosystems [21,22,23]. It has been demonstrated that DSE inoculation improves Cd tolerance to rice, decreases Cd content in roots, and governs Cd transfer from roots to shoots [24]. Moreover, association between DSEs and roots might impact the Cd tolerance that is seen in host plants. Previous investigations have identified that DSEs can alter plant root morphology and advance the growth of the root system when it colonizes the root cortex of the host plants [25,26], and improve the host’s uptake of water, mineral nutrients, and possibly favor the storage of Cd ions in the roots [21,27]. For instance, DSE has been reported to increase the root growth and impede Cd transport from roots to shoots [28]. Furthermore, an integrated analysis of the ecological functions of DSEs revealed that melanin in mycelium significantly increased the hormone concentrations and activated glutathione (GSH) metabolism, while it also influenced the plant’s subcellular distribution and chemical forms to enhance Cd tolerance [29,30,31]. For example, DSE inoculation increased the contents of indoleacetic acid (IAA), abscisic acid (ABA), and GSH in maize leaves under Cd stress [32]. Studies have found that DSEs led to the high uptake of Cd and bind Cd in the cell wall by altering the content of polysaccharide fractions [33,34]. As suggested by Xiao et al. [34], it was shown that pectin and hemi-cellulose fractions are involved in the enhancement of Cd accumulation in DSE-colonized maize cell walls. In addition, Zhan et al. [35] addressed the fact that DSE can uptake Cd and store it in cell walls in the form of insoluble phosphate–Cd complexes in DSE mycelia, which was essential for the enhanced Cd tolerance. Furthermore, the Cd–phosphate complexes can restrict the entry of Cd inside cells and alleviate the Cd-induced toxicity in plants [31]. Nevertheless, it is necessary to gather comprehensive experimental evidence validating the association between DSEs and host plant root morphology with Cd subcellular distribution and chemical forms in the tolerance of Cd stress, and arouse attention for the need to efficiently improve plant tolerance to Cd.

In the present study, we conducted a sand culture experiment with a specific DSE strain (*Exophiala pisciphila* H93) as a model DSE-association to examine the cellular mechanisms of Cd tolerance in *Zea mays*. The effects of DSE inoculation on maize growth, root morphology, and Cd content of maize planted under Cd stress (0, 5, 10, and 20 mg·kg^−1^) was investigated. We also investigated the subcellular distribution, and chemical forms of Cd taken up by the plants. We hypothesized that DSE inoculation can change the morphology of maize roots, promote the transformation of Cd into chemical forms with low activity in the root cell wall, and improve the Cd tolerance of maize.

## 2. Materials and Methods

### 2.1. Biological Materials

*Exophiala pisciphila* (accession number ACCC32496), which is highly resistant to Cd stress, was preserved in the China Agricultural Microbial Strain Conservation and Management Center (4 °C). This fungus was isolated from the roots of *Arundinella bengalensis* (Spreng.) Druce, a naturally occurring species in the lead–zinc mining area of Huize County, Yunnan Province, southwest China (103°36′ E, 26°55′ N). The DSE strain was incubated in potato dextrose agar (PDA) medium for two weeks at 28 °C to activate the strain [25].

The test maize variety was Huidan 4, which is considered to have high Cd tolerance and low Cd uptake [29]. Seeds of the same size were selected and soaked in 75% ethanol (Tianjin Damao Chemical Reagent Co., Ltd., Tianjin, China) and 10% sodium hypochlorite (10 min) for sterilization, then rinsed with sterile water (4 times). Subsequently, the corn seeds were inserted into a petri dish with water agar medium (agar 8 g∙ L^−1^) and cultivated under constant temperature at 28 °C for 3 days.

Sterilized quartz sand (Zhengjie Environmental Protection Material Co., Ltd., Zhengzhou, China) with a particle size of 0.5–1 mm was used as the culture substrate. A cylindrical symbiotic culture tube (Φ 6.5 × 40 cm) was filled with 0.4 kg of sterilized sand and 40 mL of 50% Hoagland nutrient solution, sealed, and autoclaved at 121 °C for 2 h.

### 2.2. DSE Cultivation and Pot Experiment

Eight treatments (with/without DSE + 4 levels of [Cd^2+^]) were set up in the experiment, and each treatment was replicated eight times, resulting in a total of 64 cylindrical symbiotic culture tubes (half of the replicates were used to determine biomass, root morphology, and Cd content; the other half were used to examine the subcellular and chemical form). First, a treatment without DSE was used as a control; for this, 10 g of PDA medium without the DSE strain and 2 maize seedlings were transferred to 32 tubes. For the treatment with DSE inoculation, 10 g of PDA medium containing the DSE strain and 2 maize seedlings were added to the remaining 32 tubes. All the tubes were coated with sterile AeraSeal film (150 × 150 mm) and placed in a greenhouse with a 10 h photoperiod (1000–8000 lx) at 28 °C/15 °C (daytime/night), 75% humidity for 14 days. During the growth of seedlings, the DSE strain attached to and then infected the maize roots.

CdCl_2_·2.5H_2_O (Da Mao Chemical Reagent Co., Ltd., Tianjin, China) was added to 50% Hoagland nutrient solution to achieve Cd^2+^ concentration of 0, 5, 10, and 20 mg·kg^−1^. Then, the 64 cylindrical symbiotic culture tubes were divided into four groups and supplemented with different Cd^2+^ treatments. Two similarly sized seedlings without DSE-inoculation were implanted in each culture tube for the control, while DSE-inoculated maize seedlings were used for the Cd + DSE treatments. All culture tubes were placed in a glasshouse with temperature at 28/15 °C (daytime/night) for 28 days, and distilled water was watered into each tube every 5 days until the plants were harvested. Stem and root tissues from all treatments were harvested and washed with distilled water (3 times). Then, filter paper was used to absorb residual water droplets. Plant height (from the root junction to the stem tip) was measured using a grading rule. The DSE colonization rate was determined using the grid line intersection method [36].

### 2.3. Determination of Root Morphological Indicators and Plant Biomass in Maize

Root system characteristics were scanned by an Epson Perfection V700 scanner (Seiko Epson, Nagano, Japan) to obtain WinRHIZO PRO STD4800 (Regent, Quebec, MTL, Canada) root images. The processing software analyzed the root length, surface area, average root diameter, root tip, branch, and crossing number.

The maize roots were soaked in ethylenediaminetetraacetic acid solution (EDTA, Damao Chemical Reagent Co., Ltd., Tianjin, China) for 3 h, then thoroughly washed with deionized water. The tissue was dried at 75 °C for 72 h to obtain the shoot and root biomass.

### 2.4. Determination of Cd Content in Maize

The dry samples were weighed (0.1 g) into a pressure digestion tank. Concentrated HNO_3_ and H_2_O_2_ (5:3, *v*/*v*) were added and then digested at 141 °C for 4 h. The Cd contents of shoots and roots were determined using an atomic absorption spectrophotometer (AAS ICE 3000 series, Thermo Scientific, Franklin, MA, USA).

Plant Cd uptake characteristics were calculated as: transport factor (TF) = Cd content in the aboveground part (mg·kg^−1^)/Cd content in the belowground part (mg·kg^−1^); biotransport factor (BTF) = (aboveground Cd content × aboveground biomass)/(belowground Cd content × belowground biomass)

### 2.5. Determination of the Subcellular Cd Content in Maize

Fresh samples were weighed (0.5 g) and then homogenized in prechilled extraction solution [0.25 mmol·L^−1^ sucrose + 50 mmol·L^−1^ Tris-HC1 (pH 7.5) + 1 mmol·L^−1^ DL-dithiothreitol] at a ratio of 1:20 (*w*/*v*) [35,37]. The homogenate was centrifuged at 2500× *g* for 10 min to precipitate the cell wall; the supernatant was further centrifuged at 11,000× *g* for 45 min to precipitate the organelle, and the supernatant was the soluble fraction. All operations were performed at 4 °C. The cell wall and organelle were digested with HNO_3_/HClO_4_ (3:1, *v*/*v*) in a pressure digestion tank, and the extractant was used to determine the background control. The subcellular Cd content of shoots and roots was determined by atomic absorption spectrophotometer, as described above.

### 2.6. Determination of the Chemical Form Content of Cd in Maize

Six successive forms of Cd in different chemical forms were extracted with specific solutions in the following order [31,37]. (1) 80% ethanol, extracting inorganic Cd, including nitrate/nitrite, chloride, and aminophenol Cd; (2) deionized water, extracting soluble Cd of organic acid complexes and Cd(H_2_PO_4_)_2_; (3) 1 M NaCl, extracting pectates and protein-integrated Cd; (4) 2% acetic acid (HAc), extracting insoluble phosphate Cd, including CdHPO_4_ and Cd_3_(PO_4_)_2_ and other Cd–phosphate complexes; (5) 0.6 M HCl, extracting oxalate Cd; and (6) Cd in residue.

Fresh samples (0.5 g) were homogenized in the extraction solution, diluted at a ratio of 1:50 (*w*/*v*), and shaken for 22 h at 25 °C. The homogenate was centrifuged at 5000× *g* for 10 min to obtain the first supernatant. The sediment was resuspended twice in the same extraction solution, then incubated for 2 h at 25 °C, and centrifuged at 5000× *g* for 10 min, and finally the supernatant was pooled three times. The other five extraction states were extracted using the same procedure. The supernatant of each solvent was then evaporated to constant weight on an electric plate at 70 °C and digested with an acidic oxidizing mixture of HNO_3_/HClO_4_ (3:1, *v*/*v*) at 145 °C. The Cd concentrations associated with the different chemical forms were determined via spectrophotometer.

### 2.7. Data Processing and Statistical Analysis

The experimental data are the average of 4 replicates and are expressed as the mean ± standard deviation. Microsoft Excel 2013 was used to process the experimental data. IBM SPSS 21.0 (23.0, IBM Corp., Chicago, IL, USA) was used for multivariate ANOVA, and between-group multiple comparisons using the least significant difference (LSD) model (*p* < 0.05); two-way analysis of variance (ANOVA) was used to analyze the effects of Cd stress and DSE inoculation on maize root morphology and Cd tolerance and interaction effects; Pearson’s method was used for data correlation analysis (*p* < 0.05 or *p* < 0.01). Origin (Version 9.0 Pro, OriginLab Inc., Northampton, MA, USA) was used for data plotting.

## 3. Results

### 3.1. Infection Characteristics in Maize Roots

As shown in Appendix A, under sand culture conditions, all inoculated treatments were successfully colonized by DSE, and the colonization rate increased (25.8–42.6%) significantly with increasing Cd stress. No DSE structures were observed in the roots of non-inoculated maize.

### 3.2. Maize Growth

As displayed in Figure 1, both Cd stress and DSE inoculation treatments significantly affected the maize biomass and height. Compared with the treatment without DSE inoculation, DSE inoculation significantly increased shoot biomass by 43.2% and decreased root biomass by 25.0% in the absence of Cd (0 mg·kg^−1^). DSE inoculation under Cd stress (5, 10, and 20 mg·kg^−1^ Cd) significantly increased the shoot and root biomasses of the maize. In addition, the mitigation of plant biomass reduction was more pronounced under the 20 mg·kg^−1^ Cd stress, and DSE inoculation increased the maize shoots and roots biomasses by 68.6% and 17.2%, respectively (Figure 1A).

The plant height of maize decreased by 23.5%, 35.3%, and 41.2% as the concentration of Cd increased (non-inoculated DSE), respectively. However, DSE inoculation significantly alleviated the reduction in plant height by 41.2%, 15.4%, 4.5%, and 56.0% under different Cd stress, respectively, compared to the control (Figure 1B).

In addition, the two-way ANOVA results showed that Cd stress had highly significant effects on maize root biomass and plant height, and DSE inoculation on plant biomass was commonly positive. There was a highly significant interaction between DSE inoculation and Cd stress on maize biomass and plant height (Figure 1).

### 3.3. Root Morphology

As documented by the root morphology of maize under Cd stress (Table 1), DSE inoculation caused significant changes in maize root morphology under Cd stress. Under 5 mg·kg^−1^ Cd stress, the maize root length, tips, branches, and root crossing number of DSE-inoculated plants significantly increased by 75.2%, 109.3%, 96.4%, and 221.2% relative to the control (no DSE inoculation), respectively, but the average root diameter decreased by 25.0%. Moreover, under the 10 and 20 mg·kg^−1^ Cd stress, DSE inoculation increased the root branch number by 86.3% and 24.0%, respectively, and the root crossing number significantly increased by 134.6% and 95.1% on average.

Combined with two-way analysis, the results showed that Cd stress and DSE inoculation significantly affected maize root length, surface area, tips, and crossing number, and there was a significant interaction (Table 1).

### 3.4. Cd Content and Cd Uptake by Maize Plants

In the current study, across all treatments, in both shoots and roots, the Cd content and Cd uptake in maize increased with increasing Cd stress. DSE inoculation under five milligrams per kilogram Cd stress treatment resulted in significant increases in Cd content and uptake in maize shoots by 141.3% and 180% on average, relative to non-inoculated DSE, and significant decreases in root Cd content and uptake by 40.0% and 28.0%, respectively. However, under the 20 mg·kg^−1^ Cd stress, DSE inoculation decreased the Cd content in the shoots of maize by 35.1%. In addition, it was observed that the Cd content and uptake in maize roots were higher than those in shoots (Figure 2A,B). Under 5 mg·kg^−1^ Cd stress, DSE inoculation resulted in extremely significant increases in the transport and biotransport coefficients of maize compared with no DSE inoculation, which increased by an average 203.6% and 184.4%, respectively. The transport coefficient of maize was significantly decreased by 26.3% under the 20 mg·kg^−1^ Cd stress with DSE inoculation treatment (Figure 2C,D). These results shed light on the fact that, with DSE inoculation, more Cd remained in the roots while the translocation of Cd from the roots to the shoots was restricted.

The two-way ANOVA showed that DSE inoculation and Cd stress had highly significant effects on Cd content and uptake in the shoots of maize, but the interaction between the two was not significant (Figure 2A,B). Meanwhile, inoculation with DSE and Cd stress significantly affected the transport coefficient and biological transport coefficient of maize, and there was a highly significant interaction (Figure 2C,D). The increased Cd stress stimulated the adsorption of Cd by plants, while the Cd content in the roots was increased by DSE inoculation.

### 3.5. Subcellular Distribution of Cd

The subcellular distribution of Cd in maize shoots and roots was investigated for all treatments (Figure 3). With the increase in substrate Cd stress, Cd accumulation in each subcellular fraction of shoots and roots became increasingly higher. Across all treatments, the cell wall of maize roots and shoots had the highest Cd contents and proportions, accounting for 42–66% of the total, followed by 35–50% of the distribution in the soluble fraction in the vesicles, and with a minor distribution in the organelles in both shoots and roots. Meanwhile, under 20 mg·kg^−1^ Cd stress, DSE inoculation resulted in significant increases in the Cd content and proportion in the cell walls of maize relative to the control. Among them, compared with the blank comparison group, the Cd content of shoots and roots increased by 49.0% and 84.5% on average, respectively; the proportion in shoot cell walls significantly increased from 46% to 58% (Figure 3A), and the proportion in roots increased from 54% to 66% (Figure 3A). Moreover, DSE inoculation significantly reduced the Cd content and proportion in the soluble fraction in both shoots and roots. The two-way ANOVA indicated that Cd stress and DSE inoculation treatments had interactive effects on the distribution of subcellular Cd contents in shoots and roots (Figure 3).

### 3.6. Chemical Forms of Cd

In the current study, Cd concentrations of different chemical forms in the shoot tissues were assessed (Table 2). We found that DSE inoculation caused dramatic changes in the contents and proportions of Cd chemical forms in maize shoots under Cd stress. For example, the pectates and protein-integrated Cd and insoluble phosphate Cd were mainly predominant, totaling 38.0–70.4% and 16.6–51.4% of the extracted state, respectively. With Cd stress (5, 10, 20 mg·kg^−1^), DSE inoculation resulted in significant increases of 63.8%, 45.1%, and 100.0% in insoluble phosphate Cd contents in shoots relative to the non-inoculation DSE treatment. Meanwhile, DSE inoculation significantly decreased the proportions of pectates and protein-integrated Cd in maize shoots, with a range of 16.2–22.9%. In contrast, the proportion of insoluble phosphate Cd increased significantly, with a range of 46.9–83.7% on average. In addition, a significant change was observed under 20 mg·kg^−1^ Cd treatment. Compared to the control group, DSE-inoculated maize had a significantly increased proportion of insoluble phosphate Cd in shoots by 60.6%. Moreover, the DSE-inoculated maize shoot proportions of pectates and protein-integrated Cd significantly reduced by 21.0% compared to non-inoculated maize, under 20 mg·kg^−1^ Cd stress.

The two-way ANOVA revealed highly significant interactions between Cd stress and DSE inoculation in the contents and proportions of Cd chemical forms in maize shoots (Table 2).

DSE inoculation also induced changes that were specific to tissue type (Table 3). For the roots, pectates and protein-integrated Cd were predominant in both non-inoculated and inoculated maize (30.1–57.7%), followed by insoluble phosphate Cd, which accounted for 17.9–33.3% of the total extracted state. Water-soluble Cd accounted for 12.2–28.4% of the total. Under Cd stress (5, 10, 20 mg·kg^−1^), after DSE inoculation in maize, the proportions of pectates and protein-integrated Cd in the roots significantly decreased by 32.4%, 15.6%, and 26.0% on average, respectively. Furthermore, the content (238.5%, 52.5%, and 38.0%) and proportion (238.7%, 52.5%, and 38.0%) of insoluble phosphate Cd increased significantly. Furthermore, with the increasing Cd^2+^ levels, the maize with DSE experienced a significant increase in the content and proportion of inorganic Cd in the roots (Table 3).

In the present study, the two-way ANOVA results revealed that Cd stress significantly affected the contents and proportions of chemical forms of Cd in maize roots. There was a highly significant interaction between Cd stress and DSE inoculation in the content and proportion of chemical forms in roots (Table 3).

### 3.7. Correlation Analysis

As summarized in Appendix A, the correlation analysis indicated that maize Cd content had significantly positive correlations (*p* < 0.05) with root length, tips, branches, and crossing number and with the proportions of Cd chemical forms and subcellular distribution (cell wall, soluble fractions, organelle) (*p* < 0.05). Additionally, significantly positive correlations were observed between root morphological traits and the proportions of Cd insoluble phosphate and subcellular distribution (*p* < 0.05).

## 4. Discussion

### 4.1. Effects and Mechanisms of Endophytic Root Fungi Affecting Plant Growth and Cd Content

Endophytic root fungi have significant effects on plant growth and heavy metal tolerance in host plants. When plants are stressed by harsh habitats, endophytic root fungi play a non-negligible role in mitigating the impact of the adversity through different adaptive mechanisms and improving resilience in symbiosis in plant growth and development [38,39,40]. For example, increasing evidence shows that in plants inoculated with DSE under Cd stress, the chlorophyll content increases, the photosynthetic physiology is enhanced, and the uptake of mineral nutrients and water improves [27,41]. Meanwhile, melanin in DSE mycelium has a strong antioxidant capacity, which increases the activity of antioxidant enzymes and antioxidant content in host plants [23,31,42]. Furthermore, DSE can alleviate membrane lipid peroxidation and enhance plants’ Cd tolerance [28,30]. Zhu et al. [43] found that colonization by DSE enhanced metal tolerance and improved tomato plant growth due to the reduced Cd uptake and accumulation into roots and shoots, while enhancing the activities of antioxidant enzymes to eliminate reactive oxygen species stress induced by excessive metals. Overall, these previous studies implied that DSE-inoculated roots of plants can promote plant growth. Our results indicated that under Cd stress, DSE inoculation increased maize plant biomass and decreased Cd content and uptake in maize. These results are in line with previous studies [26,29,43,44]. Likewise, DSE colonization altered metal redistribution to roots and stems to enhance metal tolerance and promoted the growth of the host plant [19,21]. Moreover, studies on *Clethra barbinervis* and *Zea mays* have shown that endophytic root fungi expand the area of the host root system through hyphae and root morphology changes, and then increase plant root biomass [45,46]. In addition, DSE will cause plants to distribute more physiological activity products to roots and stems, such as photosynthetic products, which may improve the Cd tolerance of plants [47,48]. Significantly, DSE can improve the rhizosphere environment by promoting the secretion of organic acids and other substances, which can enhance water and mineral nutrient uptake, thereby reducing the impact of Cd stress on plants [49,50].

### 4.2. Effects and Mechanisms of Endophytic Root Fungi in Regulating Root Morphology under Cd Stress

Root morphology and distribution are of crucial significance for the enhanced Cd tolerance in plants, which enhance plant adaptation to environmental stress by regulating adaptive responses, and Cd uptake may be influenced by root morphology more than root biomass [6,51]. Under Cd stress, endophytic root fungi colonization dramatically altered the morphology and structure of host plant roots (including increased root surface area, volume, tip number, and biomass) [25]. Notably, fungal colonization can promote the development of lateral and adventitious roots, improve root activity and mineral nutrients and water uptake [33,52]. The secondary metabolites synthesized and secreted by DSEs may also alter the inter-root microenvironment, directly affecting root growth and plant biomass [53]. Additionally, root morphological development is closely related to plant hormones, which promote the division and elongation of epidermal cells and make the host adapt better to the stressful environment [54,55]. This is attributed to the significant changes in the balance of host hormones, which slows down the restriction of plant growth by regulating the ratio of hormones of the DSE-inoculated plants [32,42].

Our results demonstrate that DSE inoculation significantly altered maize root morphology (root length, root tips, branches, and crossing number) and increased root biomass under Cd stress, which is generally consistent with other studies [25,29,32]. Furthermore, the Cd content of maize was significantly positively correlated with root morphology [25]. As discussed, it is possible that under Cd stress, endophytic root fungi promote the growth and elongation of root cells by regulating hormone levels, antioxidant enzymes, and other secondary metabolites in the root system [32,55,56]. Changes in physiological and metabolic processes increase root biomass and morphology, promote plant root growth, and protect plants from Cd stress [43]. Importantly, Cd uptake becomes more accessible with increasing root surface area and biomass with DSEs, possibly because DSE inoculation improves nutrient uptake and promotes plant growth under Cd stress [21,50]. It is therefore important that DSEs help roots adapt to Cd stress by regulating root morphology, improving plant tolerance to Cd, and showing a positive response to Cd stress.

### 4.3. Effects and Ecological Significance of Endophytic Root Fungal Regulation of Root Morphology on Plant Cd Uptake

The distribution of metallic ions in plant tissues is closely associated with their toxicity, and the plants cell wall is the main site for heavy metal detoxification, which is mainly composed of polysaccharides and proteins [33]. Their chemical properties are associated with negatively charged functional groups such as carboxyl, hydroxyl, and sulfhydryl groups [12]. Moreover, pectin methylesterase in the cell wall can reduce pectin methylation and release many negative charges on the carboxyl group [14], which can improve the binding capacity of Cd^2+^ and limit the translocation of Cd [57]. Thus, cell walls were considered a potential reservoir for Cd. Simultaneously, among the different extractants with the continuous enhancement of extractant polarity, the migration ability and toxicity of the extracted Cd binding forms decreased continuously in plants [58]. For instance, the complexation of Cd extracted by 1 M NaCl with organic ligands can reduce Cd ion activity and Cd toxicity [12]. Furthermore, insoluble phosphate Cd and pectates and protein-integrated Cd adsorbed on the cell wall have lower migration ability and toxicity than those in the form of inorganic salts, such as nitrate, chloride, and amino acid salts of Cd [59].

These data also demonstrate that under Cd stress, DSE inoculation increased the proportions of phosphate- and oxalate-bound Cd with low activity in maize, which is consistent with previous reports on DSE-colonized maize [26,31,35]. One explanation for this phenomenon is that Cd has a strong affinity for sulfhydryl groups in proteins or organic compounds [59], so it can result in more Cd^2+^ complexed with pectic acid and protein and remain in the cell wall, reducing the intracellular free Cd content and the toxicity of Cd to plants [16,35,60]. Furthermore, DSE-inoculated plant roots could modulate the polysaccharide components in the root cell wall, which provide binding and adsorption sites for Cd [31]. DSE-colonized maize promotes the complexation of carboxyl, hydroxyl, acidic groups, and other functional groups with Cd^2+^ so that more Cd is fixed in the root cell wall, and the continuous growth of plant cells is promoted [33,34]. In short, under Cd stress, DSE inoculation significantly affected the chemical forms of Cd in plant tissue and reduced the Cd content and uptake in plants, restricted the migration of Cd from roots to shoots, and enhanced the Cd tolerance [12,61]. Notably, DSE colonization significantly improved the compartmentalization of root cell walls. One possible reason for the increased Cd content was likely attributed to the DSE inoculation increasing cellular enzymatic activity and upregulating the expression of heavy metal ATPase genes involved in cell walls [35,56]. These processes result in reduced Cd phytotoxicity in root organelles and reduced overall Cd^2+^ translocation in the aerial parts of plants. Most importantly, melanin in DSE mycelium, a complex biomolecular compound composed of phenolic and aromatic compounds, is thought to support the structural rigidity of the cell wall, and may enhance fungal tolerance to various stressors, such as heavy metals and oxidative conditions [21,51]. This is essentially due to the fact that the exudates of endophytic fungi or exudates after symbiosis with roots can chelate Cd in soil and plants, thereby reducing the migration rate of Cd [19]. Evidently, another possible reason is that DSE inoculation improves the uptake of nutrients such as N and P, which would contribute to insoluble Cd complexation [62]. In addition, plant nutrition significantly affects the metabolism and composition of cell walls, which may make the cell wall less permeable to Cd and restrict the translocation of Cd ions into the cell [63].

The ecological function of endophytic root fungi in regulating Cd tolerance in plants is mainly related to root morphology [25]. Another important finding was that the proportion of phosphate Cd increased [42]. The root morphology was significantly correlated with the proportions of phosphate-bound Cd and cell wall Cd. These results indicate that the DSE inoculation could promote the combination of Cd with phosphate complexes in the cell wall and Cd uptake in maize roots [25,35,61]. Thus, DSE inoculation does seem to improve maize Cd tolerance and restrict Cd transport from roots to shoots. These findings further confirm the ingenious effect of endophytic root fungi on plant growth promotion and Cd transport reduction, and provide a basis for realizing crop safety in Cd-contaminated cultivated soils.

## 5. Conclusions

Under Cd stress, DSE inoculation promoted the growth of maize and changed maize root morphology. It also promoted the retention of Cd in the roots by increasing the distribution ratio between the cell wall and insoluble phosphate Cd, which reduced the transport of Cd to shoots. Correlation analysis revealed that Cd uptake and root morphology were significantly correlated with the proportions of insoluble phosphate and cell wall of Cd. We conclude that DSE inoculation changes maize root morphology and promotes the transformation of insoluble phosphate Cd and its retention in the root cell wall. This helps alleviate Cd biotoxicity, improve plant growth, and reduce Cd uptake in maize.

## Figures and Tables

**Figure 1 jof-09-00531-f001:**
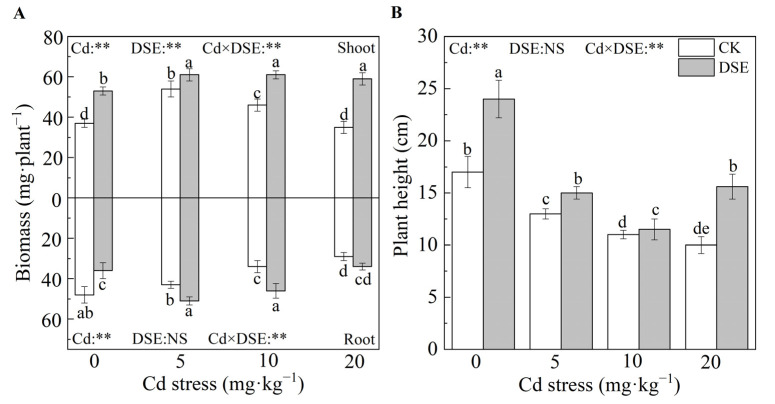
Effects of DSE inoculation on maize biomass (**A**) and height (**B**) under Cd stress. Error bar indicates standard deviation (*n* = 4). Cd: cadmium treatment, CK: non-inoculated control, DSE: *Exophiala pisciphila* inoculation. Different lower-case letters refer to *p* < 0.05 according to the LSD test. “NS” and “**” mean no significance and *p* < 0.01 according to two-way ANOVA, respectively.

**Figure 2 jof-09-00531-f002:**
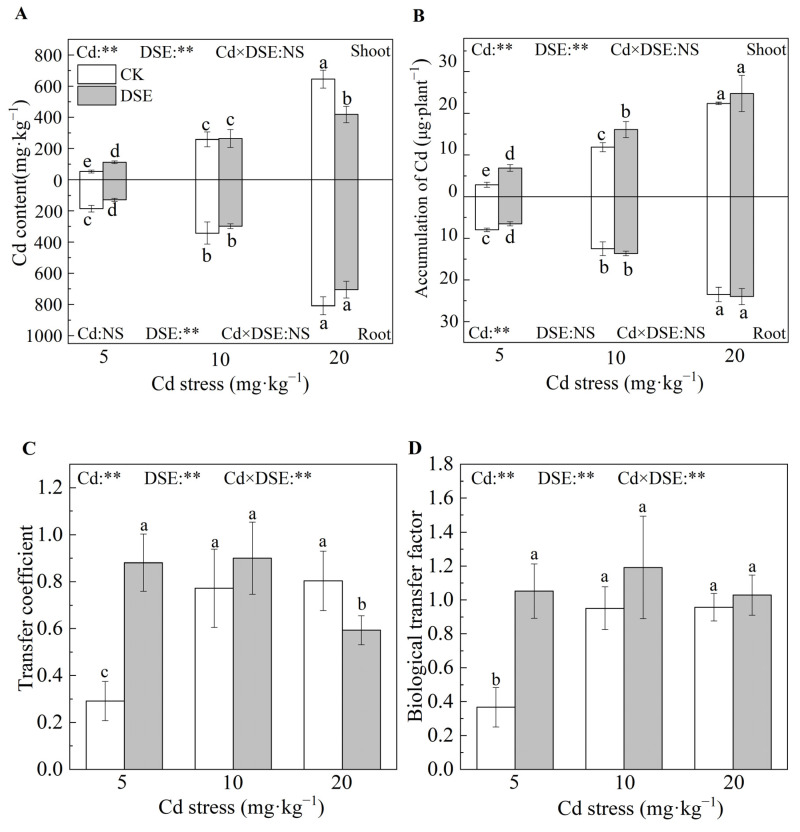
Effects of DSE inoculation on Cd content and uptake of maize. (**A**) Cd content. (**B**) Accumulation of Cd. (**C**) Transfer coefficient of maize. (**D**) Biological transfer factor of maize. Error bar indicates standard deviation (*n* = 4). Cd: cadmium treatment, CK: non-inoculated control, DSE: *Exophiala pisciphila* inoculation. Different lower-case letters refer to *p* < 0.05 according to the LSD test. “NS” and “**” mean no significance and *p* < 0.01 according to two-way ANOVA, respectively.

**Figure 3 jof-09-00531-f003:**
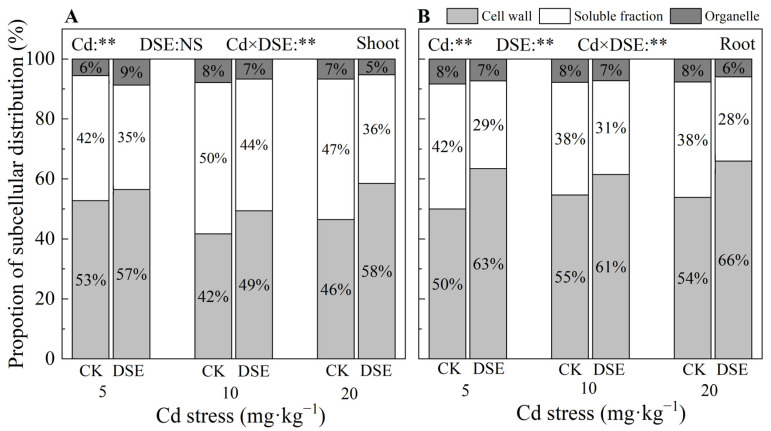
Effect of DSE on the subcellular distribution of Cd in maize shoots (**A**) and roots (**B**) under Cd stress. Cd: cadmium treatment, CK: non-inoculated control, DSE: *Exophiala pisciphila* inoculation, “NS” and “**” mean no significance and *p* < 0.01 according to two-way ANOVA, respectively.

**Table 1 jof-09-00531-t001:** Effects of DSE on root morphology of maize under Cd stress.

Cd Stress (mg·kg^−1^)	Treatment	Root Length (cm)		Root Surface Area (cm^2^)		Average Root Diameter (mm)		Root Tips Number		Root Branch Number		Root Crossing Number
0	CK	131 ± 27 b		31 ± 4.4 ab		0.76 ± 0.06 a		467 ± 84 bc		1423 ± 204 cd		53 ± 7.9 c
	DSE	165 ± 26 b		36 ± 5.4 ab		0.70 ± 0.06 a		517 ± 66 bc		1499 ± 103 c		64 ± 30 bc
5	CK	121 ± 26 b		29 ± 5.8 ab		0.76 ± 0.11 a		355 ± 72 bc		1272 ± 151 d		47 ± 27 bc
	DSE	212 ± 19 a		38 ± 4.3 a		0.57 ± 0.04 b		885 ± 94 a		2495 ± 320 a		151 ± 43 a
10	CK	113 ± 33 b		25 ± 6.0 b		0.70 ± 0.07 a		361 ± 82 c		1031 ± 128 d		49 ± 23 bc
	DSE	167 ± 28 ab		35 ± 7.0 ab		0.66 ± 0.05 ab		500 ± 73 b		1921 ± 182 b		115 ± 20 ab
20	CK	123 ± 15 bc		28 ± 3.6 b		0.73 ± 0.05 a		551 ± 61 bc		1226 ± 174 d		41 ± 10 c
	DSE	132 ± 25 b		27 ± 4.5 b		0.66 ± 0.08 ab		593 ± 87 bc		1520 ± 105 c		80 ± 17 b
Results of two-way ANOVA												
Cd	**		*		NS		*		NS		**	
DSE	**		**		**		**		**		**	
Cd × DSE	**		*		NS		**		**		**	

All values represent the means ± standard deviations, *n* = 4. Cd: cadmium; CK: non-inoculated control. DSE: *Exophiala pisciphila* inoculation. Different lower-case letters in the table refer to *p* < 0.05 according to LSD test. “NS”, “*”, and “**” mean no significance, *p* < 0.05, and *p* < 0.01 according to two-way ANOVA, respectively.

**Table 2 jof-09-00531-t002:** Effect of DSE inoculation on chemical forms of Cd contents (mg·kg^−1^) in the shoots.

Cd Stress (mg·kg^−1^)	Treatment	Inorganic Cd		Water Soluble Cd		Pectates and Protein-Integrated Cd		Insoluble Phosphate Cd		Oxalate Cd		Cd in Residue
5	CK	0.10 ± 0.01 c (4.3%)		0.18 ± 0.00 c (7.9%)		1.12 ± 0.12 dc (49.3%)		0.80 ± 0.09 d (35.0%)		0.07 ± 0.00 c (3.0%)		0.010 ± 0.00 b (0.43%)
	DSE	0.12 ± 0.04 bc (4.7%)		0.06 ± 0.01 d (2.5%)		0.97 ± 0.02 d (38.0%)		1.31 ± 0.19 b (51.4%)		0.09 ± 0.00 b (3.3%)		0.003 ± 0.00 d (0.12%)
10	CK	0.11 ± 0.04 bc (4.0%)		0.27 ± 0.02 b (9.3%)		1.37 ± 0.32 c (47.6%)		0.71 ± 0.02 d (16.6%)		0.09 ± 0.01 b (3.1%)		0.006 ± 0.00 c (0.20%)
	DSE	0.15 ± 0.01 ab (6.3%)		0.17 ± 0.02 c (7.0%)		1.34 ± 0.15 c (55.3%)		1.03 ± 0.04 c (29.0%)		0.05 ± 0.00 d (3.3%)		0.003 ± 0.00 d (0.14%)
20	CK	0.18 ± 0.02 a (3.5%)		0.28 ± 0.02 b (5.7%)		3.53 ± 0.17 a (70.4%)		0.83 ± 0.06 d (16.6%)		0.16 ± 0.00 a (3.3%)		0.027 ± 0.00 a (0.54%)
	DSE	0.08 ± 0.00 c (1.5%)		0.59 ± 0.04 a (10.9%)		3.02 ± 0.12 b (55.6%)		1.66 ± 0.08 a (30.5%)		0.07 ± 0.01 c (1.3%)		0.003 ± 0.00 e (0.05%)
Results of two-way ANOVA												
Cd	**		**		NS		**		**		**	
DSE	NS		**		**		**		**		**	
Cd × DSE	**		**		**		**		**		**	

All values represent the means ± standard deviation, *n* = 4. Cd: cadmium; CK: non-inoculated control. DSE: *Exophiala pisciphila* inoculation. Different lower-case letters in the table refer to *p* < 0.05 according to the LSD test. “NS” and “**” mean no significance and *p* < 0.01 according to two-way ANOVA, respectively. Figures in parentheses are the percent of total cadmium.

**Table 3 jof-09-00531-t003:** Effect of DSE inoculation on chemical forms of Cd contents (mg·kg^−1^) in the roots.

Cd Stress (mg·kg^−1^)	Treatment		Inorganic Cd		Water Soluble Cd		Pectates and Protein-Integrated Cd		Insoluble Phosphate Cd		Oxalate Cd		Cd in Residue
5	CK		0.16 ± 0.01 e (10.8%)		0.32 ± 0.02 e (21.5%)		0.66 ± 0.14 d (44.5%)		0.26 ± 0.03 e (17.9%)		0.07 ± 0.00 e (4.9%)		0.006 ± 0.00 b (0.39%)
	DSE		0.28 ± 0.02 d (10.4%)		0.53 ± 0.01 d (20.0%)		0.79 ± 0.04 d (30.1%)		0.88 ± 0.14bc (33.3%)		0.16 ± 0.01 c (5.9%)		0.008 ± 0.00 c (0.29%)
10	CK		0.25 ± 0.03 d (6.7%)		1.08 ± 0.04 a (28.4%)		1.46 ± 0.04 c (38.4%)		0.80 ± 0.09 c (21.1%)		0.18 ± 0.02 b (4.8%)		0.023 ± 0.00 a (0.60%)
	DSE		0.45 ± 0.03 b (11.4%)		0.86 ± 0.12 b (22.1%)		1.26 ± 0.19 c (32.4%)		1.22 ± 0.13 a (31.4%)		0.10 ± 0.00 d (2.5%)		0.010 ± 0.00 c (0.25%)
20	CK		0.34 ± 0.05 c (6.1%)		0.81 ± 0.00 b (14.4%)		3.26 ± 0.34 a (57.7%)		1.00 ± 0.18 b (17.7%)		0.22 ± 0.01 a (4%)		0.008 ± 0.00 d (0.15%)
	DSE		1.09 ± 0.04 a (19.1%)		0.70 ± 0.07 c (12.2%)		2.44 ± 0.28 b (42.7%)		1.38 ± 0.05 a (24.1%)		0.11 ± 0.01 d (1.9%)		0.003 ± 0.00 e (0.05%)
Results of two-way ANOVA													
Cd		**		**		**		**		**		**	
DSE		**		**		NS		NS		**		**	
Cd × DSE		**		**		**		**		**		**	

All values represent the means ± standard deviation, *n* = 4. Cd: cadmium; CK: non-inoculated control. DSE: *Exophiala pisciphila* inoculation. Different lower-case letters in the table refer to *p* < 0.05 according to the LSD test. “NS” and “**” mean no significance and *p* < 0.01 according to two-way ANOVA, respectively. Figures in parentheses are the percent of total cadmium.

## Data Availability

Not applicable.

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
