# Peer review of "A Dark Septate Endophyte Improves Cadmium Tolerance of Maize by Modifying Root Morphology and Promoting Cadmium Binding to the Cell Wall and Phosphate"

_jof, 2023, doi:10.3390/jof9050531_

Round 1

Reviewer 1 Report

This manuscript explained important roles of DSEs which improves Cd tolerance of maize by modifying root morphology and promoting Cd binding to the cell wall and phosphate. This study shows well-planned experimental results, and these results are thought to be of interest to readers. I believe the manuscript is suitable for publication in Jof.

Author Response

Thank you very much for your review comments on my manuscript. Every review comment from you has brought important help to my revision of this paper and future scientific research work. We had asked an English language editing company, American Journal Experts, with technical and academic writing to help us modify the language of the paper. 

Reviewer 2 Report

This is a well written paper but could have been more impactful if the authors considered in the study design, the following key aspects:

1. It is clear from the data presented here that the fungus has plant growth promoting properties however, the authors failed to make this the highlight of the paper.

(a) In page 2 (lines 62-63) you make mention of the high phenolic content of the fungal cell wall and that it plays an essential role in cadmium tolerance of the host plant. Cadmium induces oxidative stress and phenolic compounds are known to have antioxidant activity.  This therefore indicates that the phenolics may be directly involved in cadmium detoxification and as a consequence, promote plant growth and root development. Why was this connection not made in this study?

(b) in lines 341 to 347, you mention that DSE produces secondary metabolites that may lead to improved plant growth and also that the fungus can influence the plant hormone concentrations. These metabolites etc could have been easily identified through Mass spectrometry and could have greatly enhanced the narrative of this paper.

2. The metal tolerance/detoxification of the fungus could have been studied in vitro and perhaps you could have also checked (in vitro) what secondary metabolites are produced when DSE is exposed to  the metal. When exposed to the metal, does the phenolic content of the cell wall change? This would have led you to a better scientific hypothesis in my opinion.

Author Response

Thank you very much for your review comments on my manuscript. We have revised the manuscript in accordance with the reviewers’ comments. Please see the attachment.

Reviewer 3 Report

Please find comments in attached file.

Author Response

(The authors gave the same response as above.)

Round 2

Reviewer 3 Report

Author has improved manuscript according to suggestions.